

# Rapid hydration and weakening of anhydrite under stress: Implications for natural hydration in the Earth's crust and mantle

Johanna Heeb[1,2], David Healy[1], Nicholas E. Timms[2], Enrique Gomez-Rivas[3]

[1]Department of Geology & Geophysics, University of Aberdeen, Aberdeen, AB24 3UE, United Kingdom
[2]School of Earth and Planetary Sciences, Curtin University, Perth, 6102, Australia
[3]Departament de Mineralogia, Petrologia i Geologia Aplicada, Facultat de Ciències de la Terra, Universitat de Barcelona, Martí i Franquès s/n, 08028, Barcelona, Spain

*Correspondence to*: Johanna Heeb (jheeb.geo@gmail.com)

**Abstract.** Mineral hydration is an important geological process that influences the rheology and geochemistry of rocks, and
the fluid budget of the Earth's crust and mantle. Steady-state differential compaction (SSDC), dry and 'wet' tests under confining pressure, and axial stress were conducted, for the first time, to investigate the influence of triaxial stress on hydration in anhydrite-gypsum aggregates. Characterization of the samples before and after triaxial experiments were performed with optical and scanning electron microscopy, including energy dispersive spectroscopy and electron backscatter diffraction mapping. Stress-strain data reveal that samples that underwent steady state differential compaction in the presence of fluids
are ~ 14 to ~ 41 % weaker than samples deformed under 'wet' conditions. The microstructural analysis shows that there is a strong temporal and spatial connection between the geometry, distribution, and evolution of fractures and hydration products. The increasing reaction surface area in combination with pre-existing gypsum in a gypsum-bearing anhydrite rock led to rapid gypsification. The crystallographic orientations of newly formed vein-gypsum have a systematic preferred orientation for long distances along veins, beyond the grain boundaries of wall-rock anhydrite. Gypsum crystallographic orientations in {100} and
{010} are systematically and preferentially aligned parallel to the direction of maximum shear stress (45° to $\sigma_1$). Gypsum is also not always topotactically linked to the wall-rock anhydrite in the immediate vicinity. This study proposes that the selective inheritance of crystal orientations from favourably oriented wall-rock anhydrite grains for the minimization of free energy for nucleation under stress leads to the systematic preferred orientation of large new gypsum grains. A sequence is suggested for hydration under stress that requires the development of fractures accompanied by localised hydration. Hydration along
fractures with a range of apertures up to 120 µm occurred in under 6 hours. Once formed, gypsum-filled veins represent weak surfaces and are the locations of further shear fracturing, brecciation, and eventual brittle failure. These findings imply that non-hydrostatic stress has a significant influence on hydration rates and subsequent mechanical strength of rocks. This phenomenon is applicable across a wide range of geological environments in Earth's crust and upper mantle.





**Graphical abstract.**

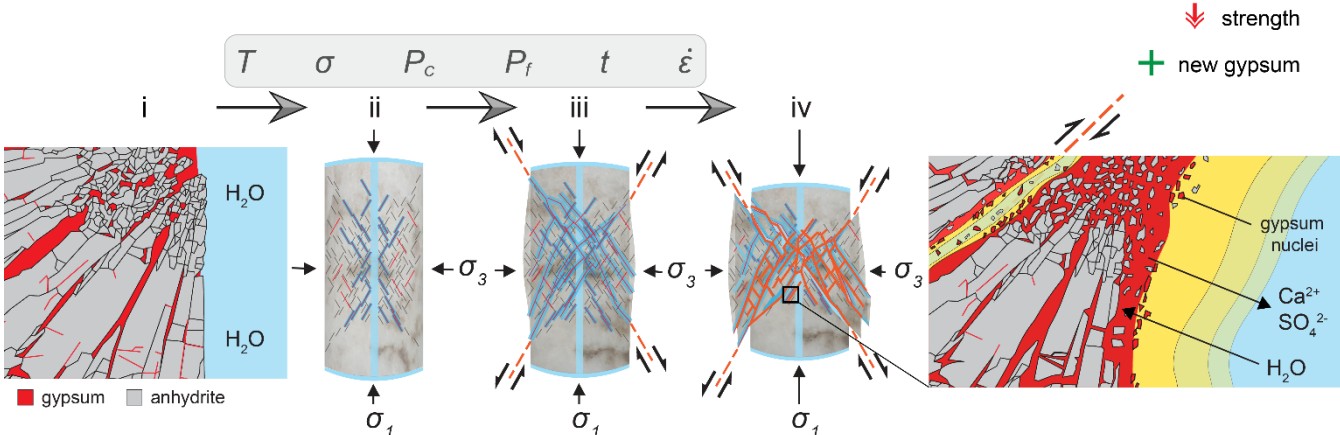





## 1 Introduction

The hydration of minerals and rocks is a common and important process in the Earth's crust and upper mantle that influences the dynamic evolution of rocks in terms of their mineral composition, fabrics, geochemistry, and rheology (e.g., Olgaard et al., 1995; De Paola et al., 2009; Llana-Fúnez et al., 2012; Leclère et al., 2018). However, hydration of rocks under non-hydrostatic

stress (rather than hydrostatic pressure) conditions has not been fully explored. Given the ubiquitous presence of non-hydrostatic stress conditions in the Earth, this represents a significant knowledge gap for an important geological process.

Hydration of anhydrite to gypsum, also called gypsification, is of interest in several economic fields, including mining, oil and gas, and storage of hydrocarbons, hazardous, and nuclear waste (e.g., Mertineit et al., 2012; Singh et al., 2018; Wang et al., 2020). It is also highly relevant in construction, as gypsum is a major cement and plaster ingredient (e.g., Farnsworth,

1925; Leininger et al., 1957; Sievert et al., 2005). Moreover, predicting anhydrite hydration is key in civil engineering, because of the potential rock volume change related to the reaction (e.g., Sass and Burbaum, 2010; Singh et al., 2018). Additionally, reactions between gypsum and anhydrite have been studied in laboratory experiments as analogues of mantle minerals (Rutter et al., 2009; Llana-Fúnez et al., 2012; Leclère et al., 2016). Due to its relevance in those fields, and because gypsification is also a very common mineral reaction in nature under surface conditions (e.g., Farnsworth, 1925; De Paola et al., 2007; Bedford,

2017) the $CaSO_4 \cdot H_2O$ system has been studied scientifically for over 90 years. Furthermore, anhydrite-bearing evaporite sequences are often the weakest horizons in sedimentary basins and form detachment horizons in foreland fold and thrust belts (e.g., Heard and Rubey, 1966; Hildyard et al., 2011). Therefore, processes that can potentially affect the mechanics of anhydrite-bearing evaporites, such as hydration, are significant because they potentially have control over the rheology and deformation behaviour of sedimentary basins and fold and thrust belts.

This study focuses on the influence of stress on hydration in the $CaSO_4 \cdot H_2O$ system (Fig. 1a), specifically the hydration of anhydrite ($CaSO_4$, orthorhombic) to gypsum ($CaSO_4 \cdot 2H_2O$, monoclinic), as an analogue for hydration systems in the Earth's crust and upper mantle. This is a simple geochemical system, and hydration is readily achievable under moderate laboratory conditions of temperature and pressure. Hydration of anhydrite under experimental differential stress conditions using natural polycrystalline rocks has been studied only recently (Li et al., 2019; Xu et al., 2019; Wang et al., 2020), with a

focus on the mechanical properties of anhydrite (Yin and Xie 2019), and the expansion or swelling associated with hydration (Serafeimidis and Anagnostou, 2013; Xu et al., 2019; Li et al., 2019). Additionally, long term (several months long) hydration experiments, mainly on powders of sieved natural and synthetic anhydrite under hydrostatic conditions (water) have failed to produce hydration products or show relatively slow hydration rates (e.g., Ramsdell and Patridge, 1929; Leininger et al., 1957; Hardie, 1967). Laboratory experiments of hydration of anhydrite under an applied non-hydrostatic stress field with focus on

microstructural evolution have not yet been attempted. Consequently, the effects of stress on hydration remain to be assessed.

The study uses a conventional triaxial deformation apparatus to investigate the rheological and microstructural response of natural anhydrite under wet and dry non-hydrostatic conditions and different strain rates. The ability to control parameters governing and influencing the reaction activity (reaction time) of hydration of anhydrite to gypsum is essential to



test the magnitude of their effects on the reactions. The following parameters were controlled: i) material-specific

characteristics (petrography) such as grain size, mineral content, and fabric; ii) experimentally controllable physical and

mechanical parameters, including temperature, fluid, effective and confining pressure, applied stress field and strain rate; and

iii) geochemical parameters like fluid composition.

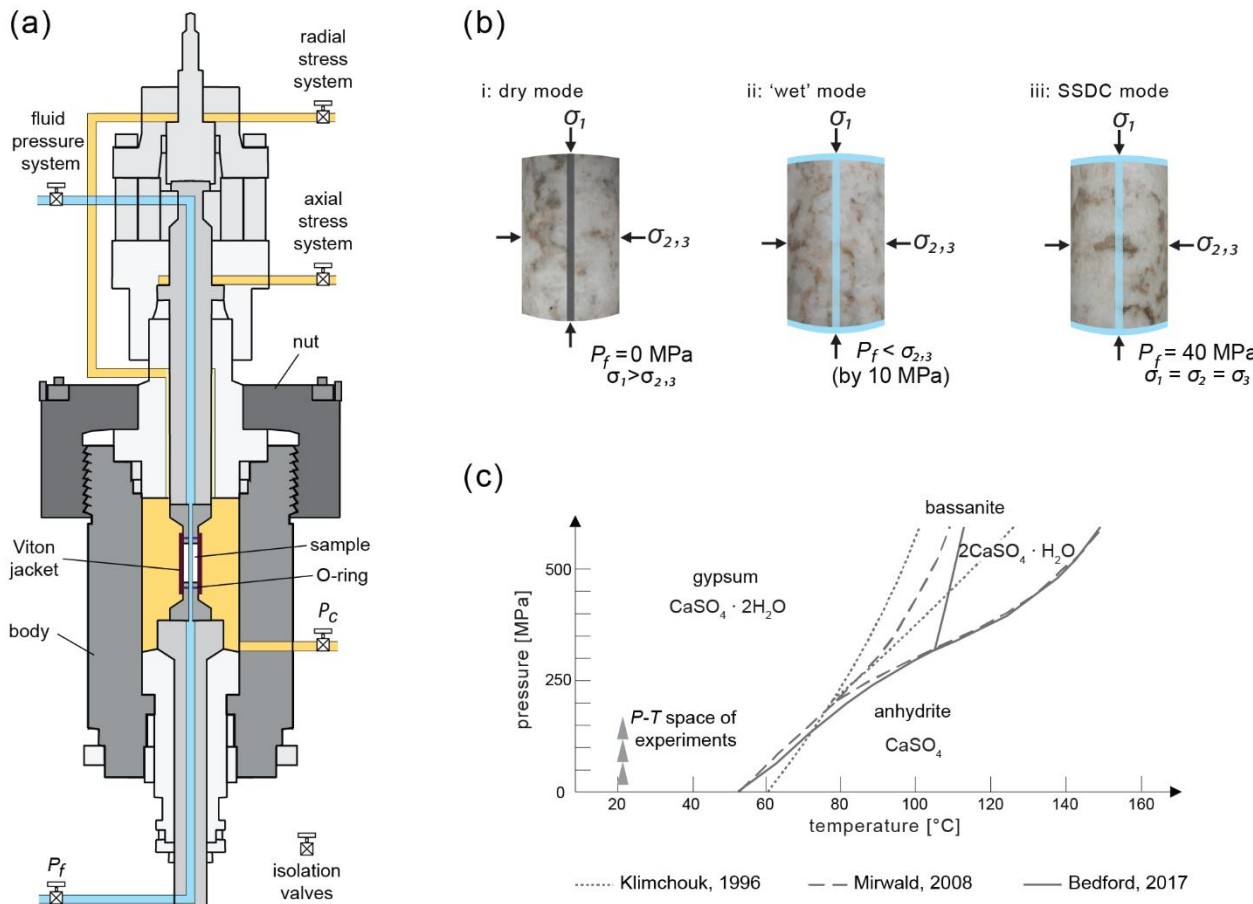

**Figure 1: Preparation and set up for triaxial experiments. a) Schematic diagram of the configuration of the triaxial rock deformation**
**apparatus (Sanchez TRI-X 250MPa/200°C). b) Experimental setup for tests. c) Phase diagram of the CaSO₄·H₂O system, adapted**
**from Klimchouk (1996), Mirwald (2008), and Bedford (2017).**

## 1.1 Review of research in the CaSO₄·H₂O system

Previous research on the interaction and evolution of stress, permeability, strength, and reaction kinetics in this chemical

system has concentrated on the dehydration reaction of gypsum (Olgaard et al., 1995; Ko et al., 1995; 1997; Wang and Wong

2003; Milsch and Scholz 2005; Milsch et al., 2011; Llana-Fúnez et al., 2012; Leclère et al., 2016). Hydration of anhydrite to

gypsum has been studied mainly on powders of sieved natural and synthetic anhydrite under hydrostatic conditions (e.g.,

Leininger et al., 1957; Hardie, 1967; Sievert et al., 2005). Hardie (1967) studied the influence of temperature on pure anhydrite



powders with different grain sizes in experiments lasting about 8 months at different temperatures between 25–60°C, without recording hydration. Only the addition of gypsum 'seeds' at similar conditions induced relatively rapid hydration. A 1:1

mixture of polycrystalline anhydrite and gypsum produced 3 % more gypsum after 83 days (Hardie, 1967).

Evolution of strength, stress-strain behaviour, permeability, and the role of grain size and fabric without any hydration or dehydration reaction in gypsum and anhydrite has been studied by Bell (1994), and De Paola et al. (2009). Bell (1994) found that anhydrite has a 'strong' unconfined compressive strength (102.9 MPa and 97.5 MPa for two types of anhydrites), whereas gypsum is ranked as having 'medium' properties (average ranges between 24.1 MPa and 34.8 MPa, depending on the pressure).

Based on the stress versus strain behaviour, Bell (1994) found that the onset of plastic deformation occurs at an earlier stage during axial loading for gypsum compared to anhydrite. Effective pressure has a significant effect on the permeability evolution under confined stress conditions and controls the brittle to ductile transition of polycrystalline, pure anhydrite during deformation (De Paola et al., 2009). During brittle failure, permeability increased dynamically by about two to three orders of magnitude. The dynamic permeability and porosity evolution during the triaxial loading tests can be summarised in three

stages: i) permeability and volume reduction through compaction is in progress, ii) permeability increase due to the onset of intra-granular micro-cracking, and iii) volume increase (dilation) and brittle failure (De Paola et al., 2009). The strength of dry anhydrite cap rock during triaxial tests increased with increasing confining pressure and slightly weakened with increasing temperature, whereas fluid contact prior to failure changed the effective pressure and lowered the strength, but not the volumetric (permeability) behaviour (Hangx et al., 2010; 2011).

**1.2 Mechanisms of anhydrite hydration**

Petrographic observations from natural rocks and experimental studies indicate that the mechanisms behind hydration (and dehydration) are (dis)solution-precipitation, and direct replacement with additional water available (Hardie, 1967; Sievert et al., 2005; Jaworska and Nowak, 2013; Bedford, 2017). Secondary gypsum is produced initially in the most fractured areas of anhydrite rocks, and forms along cracks and grain boundaries (Jaworska, 2012; Warren, 2016). Leininger et al. (1957) studied

the effect of acids, bases, and salts, particularly alkali sulphates, and showed that cations serve as activators and accelerate the hydration of gypsum, whereas anions decelerate the reaction. Activator solutions speed up the time for the appearance of maximum specific surface area and the rate of formation of maximum gypsum.

Sievert et al. (2005) developed a conceptual model for solution-precipitation that is now widely accepted (Pina, 2009; Jaworska and Nowak, 2013; Lebedev and Avilina, 2019). Hydration experiments of natural anhydrite in a ball mill with water

and activator solutions, such as $H_2SO_4$ (pH 1), 5 % $MgSO_4 \cdot 7H_2O$ and solution of calcium hydroxide, as a function of time and temperature show that the maximum specific surface area develops quickly and does not coincide with the formation rate of the maximum amount of gypsum, which takes rather longer to achieve. There is a time lag between adsorption of ions on the surface of anhydrite, which increases the specific surface area, and the formation of gypsum. Sievert et al. (2005) proposed a five step mechanism of hydration via solution-precipitation: i) rapid initial partial dissolution of $CaSO_4$ and adsorption of

hydrated $Ca^{2+}$ and $SO_4^{2-}$ ions at the surface of anhydrite; ii) slow increase of thickness of the adsorbed layer; iii) crack formation



in the adsorbed layer and counter migration of $H_2O$ (in) and $Ca^{2+}$, $SO_4^{2-}$ ions (out); iv) formation of gypsum nuclei at the surface of anhydrite and v) formation of nuclei is followed by rapid gypsum crystallization.

## 2 Materials and methods

### 2.1 Sample description and preparation

A total of eight natural anhydrite core plugs were used for the triaxial experiments. Six samples were run with water present, and two without the presence of water. The core plugs were extracted from two anhydrite-dominated surface outcrop field samples (ID prefix 'Ò') of the Òdena Gypsum Formation. This is the marginal equivalent of the salt deposits of the Cardona Saline Formation (upper Eocene) in the South Pyrenean foreland basin, Spain (Ortí Cabo et al., 1985).

Macroscopically, the Òdena samples are of a pale beige colour with discrete centimetre-scale domains that contain
light brown clay or mud inclusions (Fig. 2a). The anhydrite rocks have a minor natural gypsum content. All samples show fibro-radiate crystals of anhydrite (Fig. 2b,c). These spherulites appear either isolated or arranged in centimetre long bands. Microscopically, gypsum is located in between the anhydrite blades of the spherulites in veins (up to 10 µm in aperture), in the spherulite centres, as well as in between spherulites in broader fractures (up to 50 µm in aperture) and in the centre of the band structures. EBSD analysis shows that the crystal orientation in the spherulite 'blades' changes successively with radial
rotation, with lattice orientation being mirrored from the centre (Fig. 2d). The statistical description of the intensity of the fabric based on clustering of poles on pole figures, known as the 'multiple of uniform density' (m.u.d.) was calculated. A preferred fabric, or CPO, exists where m.u.d. > 1.

One additional core of pure gypsum was taken from an outcrop from Volterra, Italy, to compare the stress-strain behaviour and strength of anhydrite-dominated versus gypsum-dominated rocks. Volterra gypsum is a well-studied
polycrystalline material (Heard and Rubey, 1966; Ko et al., 1997; Llana-Fúnez et al., 2012), and has been used in many experiments (e.g., Olgaard et al., 1995; Hildyard et al., 2011; Brantut et al., 2012). As required for the triaxial apparatus, cores with a length (X axis) of 60 mm and a diameter of 25 mm (Y,Z dimension) were drilled out of sample blocks. Given that the sample material does not display any preferred orientation fabric on macroscale and was collected from an outcrop, cores were drilled perpendicular to bedding. The Volterra gypsum is homogeneous with no foliation, thus the orientation of the core from
this material is arbitrary.

Core plugs were drilled in the presence of water and were air-dried for 24 hours immediately afterward to mitigate any potential alteration effects. It was presumed that the exposure time to water at ambient laboratory conditions did not permit hydration of the anhydrite before deformation experiments. Pre- and post-experiment analysis of thin sections validates this assumption. A hole was then drilled (dry) into the centre of the anhydrite cores along the X axis using a drill head with a
diameter of 1.5 mm through the axis of each core to increase fluid flow and sample surface to facilitate faster and more intense hydration. All core plugs intended to be used in the experiment with fluid pressure were immersed in water and left to soak 10 minutes before starting experimental runs. Core plugs were prepared for triaxial experiments by encapsulation in Viton™





elastomer jackets to ensure a seal is formed during the experiments that shield the sample from the oil used to generate confining pressure in the cell.



**Figure 2: Macro- and microscopic sample material characterization. a) Axial orientation of cylindrical samples, whereas the long axis is defined as X and perpendicular directions are YZ (sample Ò2, pre-experiment), b) backscattered electron image, (sample Ò8,**



**post experiment), c) Crystallographic orientation EBSD map of anhydrite (sample block, initial material). Colours indicate orientation using an inverse pole figure scheme relative to map x (IPFx). Step size = 4 µm, d) Contoured equal area, lower hemisphere**
**pole figures of anhydrite data shown in c). Plots are in map x-y-z reference frame. Greyscale indicates multiples of uniform density (m.u.d.).**

All samples were analysed before and, where possible, after triaxial loading tests under confining pressure, via scanning electron microscopy using backscattered electron imaging (BSE), energy-dispersive X-ray spectroscopy (EDS), and electron backscatter diffraction (EBSD). Grain and fracture characteristics and mineral content were analysed via a range of software,
including FracPaQ (Healy et al., 2017), ImageJ (Schneider et al., 2012), and Oxford Instruments Channel 5 for EBSD data processing.

### 2.2 Microstructural characterization

Surplus material sourced from directly adjacent to the core plugs was used to prepare polished thin sections in core plug reference frame X-Y,Z and X=Y,Z-Y,Z direction before starting any experiment. Thin sections of the samples taken after
experiments were cut approximately parallel to the X axis (i.e., parallel to $\sigma_1$). Thin sections were prepared for scanning electron microscopy (SEM) via polishing with alumina, followed by a final polish with 0.6 µm colloidal silica in NaOH using a Buehler Vibromet II polisher for 2 to 4 hours. An evaporative carbon coating was applied to prevent charging during SEM. Backscattered electron (BSE) imaging was conducted with a Zeiss EVO MA10 SEM fitted with an Oxford Instruments INCA X-ray microanalysis system. A Tescan MIRA3 field emission scanning electron microscope (FE-SEM) with an Oxford
instruments electron backscatter diffraction (EBSD) acquisition system, including a Symmetry EBSD detector in John de Laeter Centre at Curtin University, was used to quantify crystallographic microstructures.

Secondary electron (SE) and BSE images were acquired, and EBSD maps with step sizes ranging from 1.7 to 50 µm were collected. Data acquisition and processing settings as well as processing procedures (Table 1) followed those of Vargas-Meleza et al. (2015) and Timms et al. (2017; 2019). Isolated, erroneous EBSD data points were removed using a 'wild spike'
correction in Channel 5, and a zero-solution infill to six nearest neighbours extrapolation was applied routinely. Misindexing of anhydrite with a range of systematic crystallographic orientation relationships was identified and data were corrected using the function in the Tango module of Channel 5.

For phase quantification, BSE images were combined with EDX phase identification data and analysed with ImageJ software (Schneider et al., 2012), using a greyscale threshold to determine phase abundance. Minor uncertainties of this
approach include greyscale variation at phase boundaries and/or due to topography of the polished surface. Additionally, fracture patterns in post-experiment sample material were quantified by manual digital tracing of gypsum-filled fractures and veins in BSE images followed by FracPaQ analysis of orientation and length of the mapped linear fracture trace segments (Healy et al., 2017).




**Table 1: Scanning electron microscopy settings and electron backscatter diffraction acquisition and processing parameters.**

| SEM | | | | |
|---|---|---|---|---|
| | Make/model | | Tescan MIRA3 FE-SEM | |
| | EBSD acquisition system | | Oxford Instruments AZtec, version 4.3/Symmetry EBSD Detector | |
| | EDX acquisition system | | Oxford Instruments AZtec, version 4.3/XMax 20 mm SDD | |
| | EBSD processing software | | Oxford Instruments Channel 5.12.72.0 | |
| | Acceleration voltage (kV) | | 20 | |
| | Working distance (mm) | | 18.5 | |
| | Tilt | | 70° | |
| **EBSD match units** | | | | |
| | Phase | Space group | $\beta$(°) | |
| | Anhydrite | Cmcm | | Hawthorne and Ferguson (1975) |
| | Gypsum | C2/c | 114.3 | Schofield et al. (1997); Boeyens and Ichhram (2002); Hildyard et al. (2009) |
| **EBSP acquisition, indexing and processing** | | | | |
| | EBSP acquisition speed (Hz) | 40 | Band detection (min/max) | 6/8 |
| | EBSP Background (frames) | 64 | Mean angular deviation (all phases) | < 1° |
| | EBSP Binning | 4 x 4 | Wild spike correction | yes |
| | EBSP Grain | high | Nearest neighbour zero solution extrapolation | 6 |
| | Hough resolution | 60 | | |

## 2.3 Triaxial deformation and hydration experimental methods

All testing was conducted with the high-pressure, high-temperature (HP/HT) triaxial rock deformation apparatus (TRI-X 250 MPa/200°C) from Sanchez Technologies at the University of Aberdeen (Fig. 1b). The parameters chosen for testing are listed in Table 2. The experiments followed three different testing modes: (i) dry; (ii) 'wet'; and (iii) steady state differential compaction (SSDC) under fluid pressure (Fig. 1c). The principal stress configuration was $\sigma_1 > \sigma_2 = \sigma_3$ throughout runs in (i) and (ii) mode and achieved through the application of a strain rate ('active' deformation). The modes (i) dry and (ii) 'wet' were created to evaluate material strength and stress versus strain behaviour for the sample material in different strain rate and pressure settings.

During 'wet' mode tests, fluid pressure was applied before initiating the strain rate. In case of steady state differential compaction under fluid pressure, the strain rate was put on hold after achieving ~ 100 MPa differential stress (75 % yield stress of the 'wet' experiments Ò5,6), to achieve microcracking and before coalescing shear fractures are supposed to have formed.





Only then was water flooded into the sample chamber and fluid pressure applied. The principal stress configuration was hydrostatic, i.e., $\sigma_1 = \sigma_2 = \sigma_3$. If failure was not achieved within 15 hours of starting SSDC the strain rate was reapplied, which reinstated the respective differential stress field. At the end of each experiment of modes (ii) and (iii) the Viton™ jackets were opened and the samples were placed in an oven at 50°C for ~ 30 minutes to prevent any further hydration from proceeding.


**Table 2: Triaxial test parameters: $\dot{\varepsilon}$ – strain rate, $P_c$ – confining pressure, $P_f$ – fluid pressure, $P_e$ – effective pressure, $t_{f.e.}$ - fluid exposure time, $t_{ssdc}$ – steady state differential compaction time, failure - stress strain curve / post-experiment core habitus, $\sigma_p$ – peak differential stress. \*Catastrophic failure after 1 hour 11 minutes during steady state differential compaction. \*\* peak stress reached during steady state differential compaction.**

| Mode | Label | $\dot{\varepsilon}$ [s$^{-1}$] | $P_c$ [MPa] | $P_f$ [MPa] | $P_e$ [MPa] | $t_{f.e.}$ [h:m] | $\sigma_p$ [MPa] | $t_{ssdc}$ [hh:mm] |
|---|---|---|---|---|---|---|---|---|
| ssdc | Ò1 | $9.7 \cdot 10^{-5}$ | 50 | 40 | 10 | 15:00* | ~ 100** | 15:00* |
| | Ò2 | $9.7 \cdot 10^{-5}$ | 50 | 40 | 10 | 06:00 | 148 | 06:00 |
| 'wet' | Ò3 | $4.4 \cdot 10^{-5}$ | 50 | 40 | 10 | 00:20 | 123 | - |
| | Ò4 | $9.7 \cdot 10^{-7}$ | 100 | 90 | 10 | 02:50 | 119 | - |
| | Ò5 | $9.7 \cdot 10^{-5}$ | 50 | 40 | 10 | 01:00 | 171 | - |
| | Ò6 | $9.7 \cdot 10^{-5}$ | 50 | 40 | 10 | 00:10 | 169 | - |
| dry | Ò7 | $9.7 \cdot 10^{-7}$ | 100 | - | 100 | - | - | - |
| | Ò8 | $9.7 \cdot 10^{-5}$ | 50 | - | 50 | - | 215 | - |
| | V | $1.0 \cdot 10^{-4}$ | 50 | - | 50 | - | 99 | - |


## 3 Results

### 3.1 Laboratory triaxial deformation tests – mechanical data

#### 3.1.1 Macroscopic sample characteristics

Brittle fractures are readily visible in the post-test cores, with different characteristics depending on the deformation mode
(Fig. 3a). All samples deformed in dry mode show bulging around the middle of the X-axis. The bulging zone shows intense fracturing via two sets of shear fractures, each with an approximate angle of 30° to $\sigma_1$. Most of the samples experienced localized failure. Samples after 'wet' testing mode show intense fracturing. The fractures follow the same pattern described for the dry samples i.e., macroscopic shear fractures. The main shear faults after SSDC are characterised by an area of intense fracturing, filled with brecciated material. The resulting lateral bulges are either not faulted or extremely faulted, compared to
the dry and 'wet' test samples. Altogether, the pieces resulting from fracturing seem smaller in size and are coated by a pale grey, soft, viscous layer.





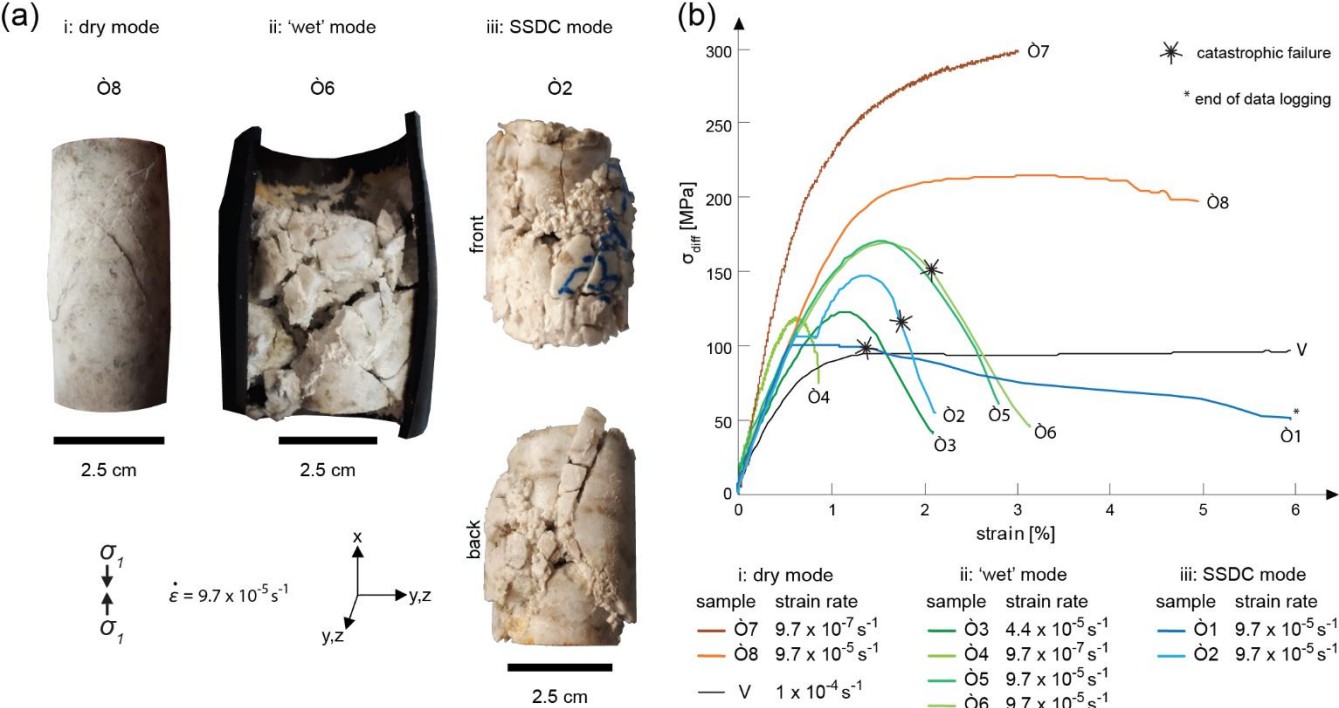

**Figure 3: Post-experimental mechanical results. a) Photographs of post-experiment cores after undergoing all three test modes. b) Stress versus strain curves, strain (%) in the shortening direction x ($\sigma_1$) on the x-axis is plotted against differential stress ($\sigma_{diff}$, axial stress/radial pressure) on the y-axis. Catastrophic failure marked for Ò1 at the point of a rapid increase of stable strain during steady state differential compaction (SSDC) phase (no strain rate applied, stable confining and fluid pressure).**

### 3.1.2 Mechanical data

The different loading modes produced distinctly different deformation behaviour, shown in differential stress versus axial strain curves (Fig. 3b). All samples show an initial phase of rapid hardening up until approximately 10 to 20 MPa differential stress. After this, total strain either stabilises or shows a minimal increase, with increasing stress. The next stage is a phase of linear elastic deformation until yield stress is reached, after which the differential stress decreases. Loading after yield stress results in different behaviour, depending on the test mode.

Dry tests showed either strain hardening (sample Ò7), or a phase of constant differential stress with increasing strain, and with increasing tendency to slight weakening (sample Ò8). The Volterra gypsum is considerably weaker compared to all anhydrite tests. The linear elastic response is limited to stresses and strains below 40 MPa and 0.25 %, respectively. The stress-strain relationship of the dry tests shows neither strain hardening nor softening and is without any sign of failure during the ongoing test. The 'wet' tests show considerably weaker behaviour compared to the dry tests. Strain weakening or softening was displayed after reaching peak differential strength (Table 2). The 'wet' experiments were stopped when steep catastrophic strain weakening happens.





The SSDC experiments behaved similarly to 'wet' and dry experiments during the first stages until strain rate was set to 0 (constant) before yield point is reached (~ 100–110 MPa) and fluid pressure was applied (20–90 MPa, Table 2) in under one minute. Sample Ò1 was stable with increasing strain for about 1 hour, before catastrophic failure at 1.35 % strain and 99 MPa differential stress. Catastrophic failure occurred at higher differential stress and lower strain conditions than when 'wet' same condition tests and Ò2 showed steep catastrophic strain weakening. During the SSDC phase, strain increases, and the

stress conditions were stable for sample Ò2. Compared with samples Ò5 and Ò6, which were run with the same strain rate, Ò2 is weaker and differential strain decreases in a steeper trend.

## 3.2 Microstructures

### 3.2.1 Fracture and gypsum-filled vein pattern analysis

A fracture pattern was analysed for gypsum-filled veins from BSE images of a thin section from 'wet' mode sample Ò3. This

sample failed with one main shear fracture (Fig. 4), which left enough solid material for detailed analysis of a 'wet' mode sample. Mapping of gypsum-filled veins in a part of the sample that features a significant vein system yielded a representative dataset for orientation analysis of all gypsum veins in view with apertures > 25 µm and of a sufficient dataset of identifiable < 25 µm wide narrow gypsum-filled veins.

Orientation analysis of all gypsum-filled fracture segments in 2D shows a preferred orientation with a prominent peak

close to 30° from the core axis (and therefore to $\sigma_1$) of all aperture classes (Fig. 4c). The wider, less abundant cracks and gypsum-filled veins show stronger preferred orientations than narrower cracks/veins or those observed in the pre-experiment undeformed Òdena anhydrite. The preferred orientation of > 25 µm gypsum-filled veins is like that of shear and extensional fracture orientations predicted by the orientation of the applied stress field during the experiment: macroscopic fractures visible in this thin section that were created by the triaxial test should have azimuths of either 30°/210° or 150°/330° relative to X, the

direction of the principal stress $\sigma_1$ (Fig. 4). However, gypsum infill implies an extensional component to the kinematics of these structures (extensional- or hybrid-shear). In detail, there are two different preferred orientations dominant in fracture populations of different widths. Veins narrower than 25 µm are almost evenly distributed around 1 % for all directions with the exception of a distinct peak around 45° counter-clockwise from X (Fig. 4c). This peak coincides with the trend of cleavage in a large anhydrite grain that dominates the lower part of the map.

Analysis of gypsum-filled vein segments with widths in the ranges of 25–50 µm, 50–100 µm, and > 100 µm show that the preferred orientation gets stronger with increasing width of the veins (standard deviation of circular mean decreases, whereas resultant increases with increasing width) (Fig. 4c). Furthermore, segment traces are longer (average segment lengths for the ranges increases from 48.34 µm to 74.43 µm to 102.11 µm) with increasing vein width.





Figure 4: Distribution of gypsum veins in sample Ò3 after 'wet' experimental run. a) BSE image showing the distribution of phases. b) Map of gypsum-filled veins, with segments coloured for orientation and line width representing vein widths (FracPaQ; Healy et al., 2017). Not all fractures smaller 25 µm are traced due to their high abundance. c) Length-weighted segment orientation rose diagrams corresponding to the dataset shown in b), with 5° bin size.





### 3.2.2 Crystallographic orientation analysis of newly formed gypsum

Crystallographic orientation mapping was performed for anhydrite and gypsum of the same area of 'wet' mode Ò3 sample from Fig. 4 (Fig. 5). The dominant form of anhydrite in the upper part of the map are spherulites comprising radially oriented anhydrite blades that progressively change their crystallographic orientation (Fig. 5a).



**Figure 5: Electron backscatter diffraction analysis of the same area shown in Fig. 4 from sample Ò3, deformed in 'wet' testing mode.**
**a) Crystallographic orientation EBSD map showing anhydrite orientations via inverse pole figure colour scheme relative to map x (IPFx). Step size = 2.2 µm. Underlying band contrast image. b) Crystallographic orientation EBSD map of gypsum with IPFx colour scheme. Enlarged insets (I) and (II) from (a) and (b), respectively, to compare crystallographic orientations of host anhydrite with vein-hosted gypsum. c), d), and e) Contoured equal area, lower hemispheres pole figures of anhydrite and gypsum for data shown in (a) and (b), respectively. Plots are in map x-y-z reference frame. Greyscale indicates multiples of uniform density (m.u.d.).**

The spherulites have an approximate diameter of ~ 700 to 1250 µm. Clusters of blocky anhydrite crystals with approximate diameters in the range of 70 to 350 µm are scattered between the anhydrite spherulites (Fig. 5a). The third fabric component is made up of large, strained crystals (1000 µm long) with cleavage, dominating the lower part of the map and visible in green colours in the EBSD crystallographic orientation map (Fig. 5a).

Anhydrite in the mapped area shows a strong CPO with the pole to {010} orientated ~ 40° counter-clockwise from X
(Fig. 5c). This fabric is dominated by aligned (cleaved) components of the large crystals, whereas the crystallographic orientations of the blocky grains are randomly oriented (Fig. 5a). The majority of the gypsum present in the mapped area is concentrated in the main vein structure (Fig. 5b). Only a small proportion of the gypsum is distributed in 'traces' inside the anhydrite fabrics. Orientation mapping shows that the gypsum filling the main veins forms domains (grains) up to ~ 1000 µm long sections have a similar crystallographic orientation (Fig. 5b I,II). Only a small fraction of crystals shows different
crystallographic orientations. However, the EBSD map shows that, locally, the sizes and spatial positions of gypsum grains in the veins do not have any relationship with the neighbouring anhydrite in the wall rock (Fig. 5b I,II). Nevertheless, pole figures show that poles to {010} of anhydrite and poles to {100} of gypsum show broad alignment (Fig. 5c,d). Similarly, poles to {001} of anhydrite and poles to {010} of gypsum tend to align in some parts of the veins (Fig. 5c,d). Overall, there is no clear link between crystallographic orientation of vein gypsum and the orientation of principal stress $\sigma_1$, or predicted shear fracture
planes. However, there is a clustering of poles to {100} and {010} in gypsum at approximately 45° to $\sigma_1$, which is parallel to the direction of maximum shear stress (Fig. 5c,d).

### 3.2.3 Characterization of fractures after steady state differential compaction

The fabric elements and phase abundance related to SSDC followed by failure are analysed from a BSE image of one of the main shear planes of sample Ò2 (Fig. 6). The thin section of this sample provides the opportunity to study gypsification related
to shear fractures after SSDC. Five domains (A to E) are defined mostly after the phase abundance contrast. In detail, defining the A/B boundary is made by compromising between abundance and fabric characteristics. The B/C boundary is easily placed by tracing a fault plane. The C/D boundary is defined mainly by the compaction contrast between domains. The D/E boundary results from a combination of fault horizon and material abundance.

Domain A has mostly blades of anhydrite with sharp edges, the spherulitic structures are still visible and gypsum is
located interstitially between these blades. The anhydrite grains are blocky towards the domain boundary, with edges that range from sharp but most commonly are rounded. There is no evidence of rotation of grains in these domains due to the kinematics of the experiment.







**Figure 6: Analysis of a shear fracture in sample Ò2 after steady state differential compaction and failure. The area in the image shows the main shear fracture that divides the lower, intact end piece of the sample core from an intact side slab. a) Backscatter electron image with domains (A – D) defined by texture and composition. Dolomite is identified based on habitus and experience from EDX results of other areas in the thin section. b) Greyscale threshold settings defined to quantify % area of phases from the backscatter image analysis via ImageJ. c) Bar chart to show % area of phases in domains and mean values of the pre-test Òdena anhydrite (same thresholds applied).**

Domain B is dominated by gypsum with a mosaic of isolated anhydrite grains (inclusions). Anhydrite is mostly rounded, some with evidence of rotation with respect to one another. The abundance of gypsum increases towards domain C, forming a layer of pure gypsum. Domain C mainly consists of clasts that contain anhydrite, gypsum, or both, and with no significant matrix. The size (long axis) of the gypsum clasts ranges from < 1 µm to > 100 µm. The big gypsum clasts can be highly fractured, with sporadic smaller anhydrite grains at the rims or as ~ 1 µm small inclusions. Almost half of the domain is porous, and gypsum content is higher than that of anhydrite. In domain D, the anhydrite grains are rotated, and embedded into a gypsum matrix. The edges are round to semi-round in shape and the particle size is up to 25 µm (length of long axis). The domain is highly brecciated with contact between particles. The boundary to domain D is defined by a series of fractures. The initial fabric is preserved in domain E but highly affected, showing abundant intra- and inter-granular fracturing. Inter-granular fractures are mostly filled with gypsum, whereas intra-granular fractures are predominantly empty. The shape of the edges of the anhydrite grains ranges from sharp to slightly rounded. Abundance analysis results are that more than half of the domain consists of anhydrite.

## 4 Discussion

### 4.1 Evidence for new gypsum formation

The strongest evidence for successful hydration and formation of gypsum is represented by the breccia vein shown in Fig. 6. The main vein has an orientation of 37.5° to X ($\sigma_1$), which is consistent with a shear fracture caused by the SSDC mode experiment. Optical assessment and greyscale threshold analysis shows that the gypsum content in and around the shear fracture is significantly higher compared to the initial sample material (Fig. 2b). The higher abundance of gypsum and rounded, rotated anhydrite grains in the margins (domains B, D) of the breccia vein are evidence for active (syn-experiment) gypsification. The centre of the breccia vein (Fig. 6, domain C) contains > 100 µm gypsum clasts, which is orders of magnitude larger than any observed pre-experiment gypsum, located in centres of anhydrite spherulites and short narrow (< 50 µm) veins (Fig. 2b). These clasts can contain small anhydrite inclusions and are derived from newly formed gypsum (Fig. 6a). Based on the distribution of the anhydrite inclusions at the margins of the gypsum clasts, the gypsum was part of a shear interface with active gypsification before brecciation occurred.

The formation of the gypsum vein system from sample Ò3, documented after a 'wet' mode experiment (Fig. 4a,b) is consistent with syn-experiment gypsification and deformation. The wide vein apertures (>> 50 µm) in combination with the systematic orientation and length of the gypsum-filled vein system of > 2.5 cm were not present in the primary sample material.



These are strong indicators for experimentally induced extension and formation of new gypsum. The wide gypsum-filled vein system formed by linked extensional fractures with a minor shear component that progressively coalesced to result in a stepped shear fracture. (Fig. 4). Additionally, the crystallographic orientation of the vein gypsum is such that poles to {010} generally

coincide with the direction of maximum shear stress during the experiments. This geometric link between gypsum growth and stress during an experiment and independent of the surrounding anhydrite has not been described before and requires further discussion.

### 4.2 Evolution and mechanisms of hydration

#### 4.2.1 Rapid hydration of anhydrite under stress

A significant outcome of this study is that hydration of anhydrite to gypsum was achieved under non-hydrostatic stress conditions over a few hours. The SSDC experiment with sample Ò2 lasted for 6 hours and produced gypsum in the fracture-related pore space created during the experiment. Sample Ò3 shows a significant amount of new gypsum in veins even after a twenty minute long 'wet' mode experiment. These results contrast starkly to previous attempts to hydrate anhydrite, which failed to produce gypsum over many months under hydrostatic conditions (e.g., Ramsdell and Patridge, 1929, Leininger et al.,

1957; Hardie, 1967). This suggests that there is an intrinsic link (or links) between the application of a non-hydrostatic stress field and the rate of the hydration reaction.

#### 4.2.2 Spatial distribution and timing relationships

Microstructural observations (Fig. 4, 5 and 6) show a paragenesis that links to the stress-strain evolution. A model to establish the spatial distribution and timing relationships of hydration products and fracture pattern development results from

experimental observations was developed (Fig. 7). During macroscopically quasi-elastic stress-strain behaviour, the onset of intragranular fracturing concentrated in the centre of the core and the orientation of shear planes (30° angle to $\sigma_1$) significantly increased sample permeability and provides three-dimensional fracture networks as pathways for fluids (Fig. 7a i). Application of fluid pressure during SSDC and 'wet' mode experiments ensured the fast distribution of $H_2O$ through these networks (Fig. 7a ii). At fracture-fluid interfaces, the presence of anhydrite, gypsum and $H_2O$ led to in situ hydration and gypsum vein

formation. Sample Ò2 had 6 hours of contact with $H_2O$ in total. Five hours and fifty-six minutes under isotropic principal stress conditions (i.e., $\sigma_1 = \sigma_2 = \sigma_3$), and less than 2 minutes from re-application of strain rate to maximum differential stress ($\sigma_{max}$).







**Figure 7: Interpretation of fracture formation and fluid distribution in the sample cores throughout triaxial tests. a) Schematic**
**fracture formation. Not all stages apply to all tests, depending on the experimental mode. b) Relation of a) to steady state differential**
**compaction stress-strain curve of sample Ò2.**

The margins of gypsum grains and large gypsum clasts contained in the brecciated zone of the shear fractures after SSDC in

sample Ò2 exceeded the gypsum formation documented after the 'wet' mode experiment in sample Ò3. Combined with the

timeline, this larger gypsum grain content strongly indicates early inter-granular fracturing combined with the formation of

new gypsum before reaching maximum differential stress. After maximum differential stress and prior to dynamic hydration

related brecciation (Fig. 7 iii), bulging and (faster) shortening of the sample in the X direction through the activation of shear

plane fractures and local extensional operation of a three-dimensional fluid pathway network occurred within two minutes.

Shearing along the main shear fractures results in rapid shortening in the X direction during the last stage (Fig. 7a iv) and is

characterized by a rapid stress drop (-10 MPa every three seconds) with ongoing strain. The onset of such catastrophic failure

thirty seconds after maximum differential stress was reached, led to the formation of cataclastic zones and brecciated veins

(Fig. 6).





### 4.2.3 Crystallographic orientation of newly formed gypsum

The crystallographic orientations of newly formed gypsum in the veins have a systematic preferred orientation for long distances along veins, beyond the grain boundaries of wall-rock anhydrite (Fig. 5a,b). Gypsum is not always topotactically
linked to the wall-rock anhydrite in the immediate vicinity, indicating that inheritance of crystal orientation from anhydrite did not lead to the strong clustering of poles. There is also no evidence of alignment of crystals with respect to the vein walls, or evidence of gypsum crystals that grew from the vein margin to its centre, and so alignment by competitive crystal growth of gypsum into the vein is unlikely. Instead, gypsum crystallographic orientations are observed to be systematically and preferentially aligned parallel to the direction of maximum shear stress (Fig. 5c). This study proposes that inheritance of crystal
orientations from wall-rock anhydrite grains combined with crystal orientations favourable for nucleation and growth under the applied stress field (e.g., stress-related minimisation of the energy barrier for nucleation) led to selective crystallographic orientations of large new gypsum grains.

### 4.3 Mechanical-chemical coupling

The spatial link between newly formed gypsum and fractures shows that hydration predominantly progressed through the
fracture network rather than a front that progressed through the sample, similar to that reported for gypsum dehydration and anhydritization (Wang and Wong, 2003; Llana-Fúnez et al., 2012). A concept for the hydration mechanism of anhydrite particles developed by Sievert et al. (2005) involves dissolution and precipitation, which was adapted here to explain hydration of the Òdena anhydrite under stress (Fig. 8). The 'wet' mode experiments make $H_2O$ groups available to new mineral interfaces during the initial intra-granular fracturing. Upon the contact of anhydrite surfaces with water, $CaSO_4$ solution and the surface
absorption layer of hydrated $Ca^{2+}$ and $SO_4^{2-}$ ions formed (Fig. 8) (Sievert et al., 2005). The increase of thickness of the absorbed layer is reportedly a slow process and needs to be followed by the crack formation in the absorbed layer and counter migration of $H_2O$ and $Ca^{2+}$ as well as $SO_4^{2-}$ ions (Sievert et al., 2005).

Pre-existing gypsum in the samples acted as a natural seeding material, which has been demonstrated elsewhere to enable (or speed up) the hydration reaction process because the kinetically challenging process of forming nuclei (e.g., Hardie,
1967; Wheeler, 1991; Sievert et al., 2005) is skipped. The enhancement of mineral replacement reactions by the presence of seeds is also a common phenomenon in other diagenetic processes, such as dolomitization (e.g., Whitaker and Xiao, 2010). Therefore, hydration was possible as soon as the samples had water contact and more likely in SSDC experiments due to the amount of time of contact with $H_2O$. However, the importance of this process is difficult to reconcile with the distinct microstructural location of new gypsum in newly formed veins, or the lack of gypsum in hydrostatic experiments. Rounded
anhydrite inclusions in gypsum margins of shear fractures and as clasts in brecciated veins (Fig. 6a) are specific indicators for the dissolution of anhydrite.



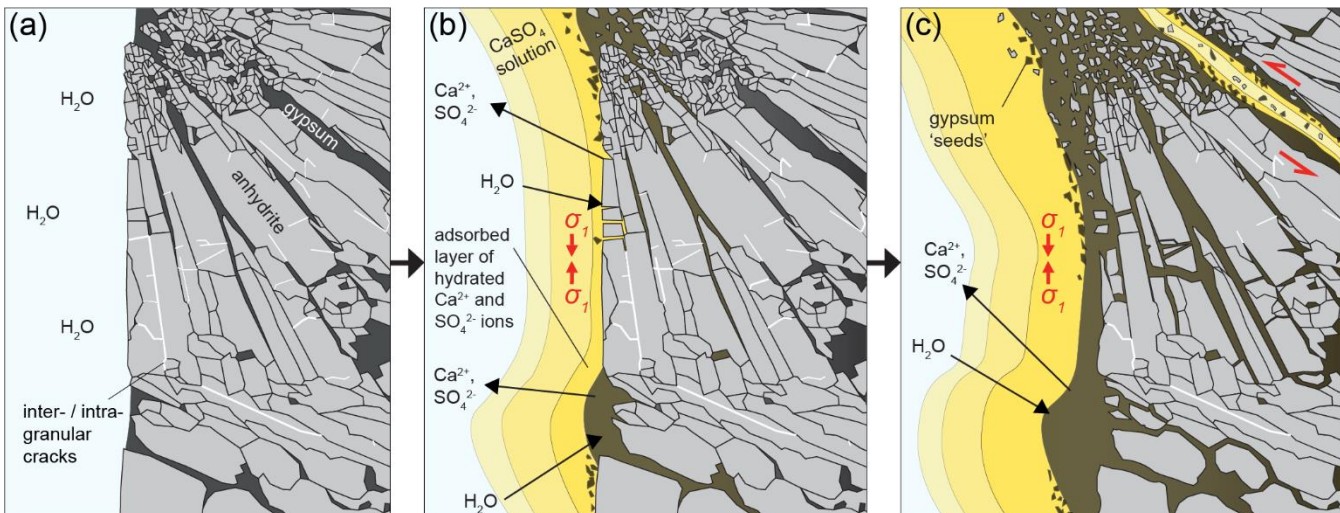

**Figure 8: Model for solution – precipitation hydration in Òdena anhydrite based on the hydration mechanism suggested by Sievert et al. (2005). The model includes a spherulite structure, cleavage, and blocky anhydrite areas in contact with water. Initial gypsum is located in veins along grain boundaries and the centre of the spherulite.**

The role of fractures is threefold: Firstly, they provide new surface area available for reaction. Secondly, they facilitate fluid flow to enable a readily available medium ($H_2O$) for solution transfer of $Ca^{2+}$ and $SO_4^{2-}$ ions. Thirdly, locally variable stresses associated with fracture propagation gave rise to spatial variations in chemical potential and as a consequence, chemical disequilibrium (Llana-Fúnez et al., 2012; Wheeler, 2018). Solid-fluid contacts will be at the pressure of the fluids ($P_f$), whilst solid-solid contacts will have a higher average normal stress, depending on the bulk effective pressure and contact area (Llana-Fúnez et al., 2012). That provides different pathways of $Ca^{2+}$ and $SO_4^{2-}$ ions during the reaction. Therefore, anhydrite solution was preferentially formed in the stressed anhydrite at fracture tips, grain boundaries, and at gypsum-anhydrite contacts. Once gypsum nuclei were established, growth was likely to be rapid, following the findings of Sievert et al. (2005).

The transformation of anhydrite to gypsum requires a significant change in volume of solid material (i. e. swelling). Upon contact with water gypsum is no longer solid but partly dissolved and starts to moderately swell (Fig. 8b). Simultaneously, anhydrite dissolution occurs and transfer of $Ca^{2+}$ and $SO_4^{2-}$ ions and $H_2O$ molecules permeate through the gypsum (Fig. 8b,c). The consumption of water acts to lower local fluid pressure, whereas replacement of anhydrite by gypsum causes swelling, counteracting the decrease in local fluid pressure. Recall that the tests were conducted at a constant *bulk* fluid pressure (held at 10 MPa lower than confining pressure) without any induced sample-scale fluid flow. Nevertheless, fresh supply of $H_2O$ at the scale of grains, pores and cracks was facilitated by the opening of a connective network of new intergranular fractures (Fig. 7a iii). Fracturing combined with the availability of water for the formation of gypsum facilitate dilatancy, which is seen as bulging of the jacketed sample charges (Fig. 7a iii). Swelling (volume increase) and water loss through $H_2O$ groups being bound into the gypsum impact activity of hydration in places. Swelling can seal up cracks and trap free water. This potentially stops the hydration reaction in places, while the water migrates into other, harder to reach environments, like grain boundaries, and facilitating hydration there with fewer $H_2O$ groups available.



Cataclastic flow and the full development of major shear fractures (Fig. 7a,b, iv) occurred after the peak stress was reached. The 'wet' tests show that these major shear fractures with thin interconnected parallel fractures and areas of wide fractures are all filled with gypsum. These form planar zones of weakness for catastrophic shear failure. For the phase after peak stress is achieved, De Paola et al. (2009) recorded a rapid increase in permeability that becomes 'chaotic' in the final

stage of failure. This is likely to be coupled with a rapid increase in the area of available reaction surfaces. The macroscopic observations show that the sample cores after experiments with applied fluid pressure, if not failed catastrophically, comprise fragmented debris of centimetre to millimetre size, covered with a white slurry. This indicates that rapid gypsum formation may occur during the last stage (only seconds long) and upon failure. The lower peak stress of sample Ò2 after re-initiation of a strain rate of $9.7 \cdot 10^{-5}$ s$^{-1}$ can be explained by the development of weakening zones due to the hydration of gypsum and filling

of cracks. Only sample Ò1 failed during SSDC. This could be due to a favourable orientation of a pre-existing zones of weakness. There is gypsum in the initial sample, in short (< 1 cm) veins with an aperture of < 50 µm. The formation of new gypsum is linked to sample failure.

### 4.3.1 Mechanical strength

A consequence of hydration under stress is the weakening of the mechanical strength during deformation. Samples Ò1 & Ò2

that experienced SSDC, have considerably lower peak strength compared to 'wet' and dry runs with the same strain rate of $9.7 \cdot 10^{-5}$ s$^{-1}$. Slower strain rates (Ò3,4,7) generate weaker peak strengths. Besides strain rate, the testing mode has the most significant influence on peak differential stress. Sample Ò8 showed the highest peak differential stress (215 MPa), and 'wet' experiments Ò5 & Ò6 were intermediate (~ 170 MPa). Sample Ò1 failed catastrophically at the beginning of the SSDC phase, with a maximum differential stress before failure of ~ 100 MPa, and therefore about 41 % less compared to 'wet' experiments.

Sample Ò2 reached a peak strength (147 MPa) after reapplication of the strain rate. The peak strength of sample Ò2 is 14 % lower than that of the 'wet' experiments.

The microstructural analysis shows that the new gypsum is located along fractures in extensional and shear orientations, creating planes of weakness, and lowering the bulk mechanical strength. A stronger connected shear fracture network developed until the onset of isotropic principal stress conditions (i.e., $\sigma_1 = \sigma_2 = \sigma_3$) likely caused the more rapidly

developed connective fracture network in samples Ò1 and Ò2. The coalescence of fractures accompanied by hydration in sample Ò1 occurred within 71 minutes under isotropic confining stress conditions once fluids were introduced.

### 4.4 The influence of stress on chemical reactions

The magnitude and significance of differential stresses that may be induced through hydration reactions of mineral systems accompanied by a solid-volume increase are poorly understood. However, hydration reactions are commonly associated with

deformation reactions like dilatant fracturing, which increases the fluid permeability. Plümper et al. (2022) showed that the hydration reaction $MgO + H_2O = Mg(OH)_2$ can induce stresses of several hundred megapascals in nature, with maximum local stresses up to ~ 1.5 GPa. This is in agreement with the findings of this study, based on which stress, as opposed to pressure,





influences the hydration reaction specifically through microcracking and faulting providing pathways for fluids and surface area for the reaction.

There are two different stress–material interactions to consider for understanding the impact of stress on chemical reactions (Wheeler, 2018). Normal stress (anisotropy) along grain interfaces and between interfaces with different orientations has the main impact on chemical reactions in the Earth, and thus, plays the key role in quantifying stress-related chemical processes (Wheeler, 2014; 2018). Chemical potential depends on a "weighted" mean stress, which means that the magnitude and orientation of stress have a relatively minor impact (Wheeler, 2018). Experiments show that narrow aqueous or other films

along (grain) boundaries may persist, even if normal stress is greater than fluid pressure (Hickman and Evans, 1995, Israelachvili, 2011). They are regarded as stressed solids rather than fluids (e.g., Israelachvili, 1992; Wheeler, 2018), which provide fast diffusion pathways (Rutter, 1976). Integral parameters for models are the grain boundary structure, assumptions about the mobility of specific components, and reaction activity (Wheeler, 2018). These include grain boundary film properties like the connection between surface and interface energies and film structure (Hickman and Evans, 1995), and the relationship

of fluid film thickness to normal stress (Israelachvili, 2011). The basic concept is that grain boundaries, representing a small-scale volume, are locally buffered by (i.e., are in local equilibrium with) the adjacent solids (Wheeler, 2018). Wheeler (2018) states that diffusion is the main mechanism of stress-related chemical processes and is active along long-range chemical reaction pathways that are provided by interconnected interfaces under crustal conditions. It is established that diffusion rates along interfaces such as grain boundaries are several orders of magnitude faster compared to intracrystalline diffusion (Dohmen

and Milke, 2010).

     Further, segregation of (incompatible) elements and their enrichment in grain interfaces is considered to have a significant impact on the physical and chemical properties of mantle rocks (Hiraga et al., 2007). Interfacial segregation linked with grain boundary character distribution (GBCD) may lead to grain boundary energy minimization (Tacchetto et al., 2021). It follows that interfacial segregation potentially influences if and where diffusion is active or accelerated in natural samples

during hydration.

     Macroscopically, the difference between the 'wet' and SSDC mode samples is that the main shear faults after SSDC are characterised by an area of intense fracturing, filled with brecciated material. The resulting lateral cheeks are either not faulted or extremely faulted, compared to the dry and 'wet' test samples. Altogether, the pieces resulting from fracturing seem smaller in size and are coated by a pale grey, soft, viscous layer. Combined with the relative mechanical weakness of samples

after SSDC compared to dry and especially 'wet' tests, it is possible to conclude that there is a difference between reaction rates and mechanisms for hydrostatic and non-hydrostatic loadings. The possible relation between the spatial distribution and crystallographic orientation of the new gypsum in veins with orientations of 45° from the maximum shear stress presented by this study are a start to understand hydration processes on the grain-pore-crack scale and how they are linked to mechanic, thermodynamic, and kinetic.





## 4.5 Implications for the Earth's crust and mantle

The main implication of this study of hydration under stress in the crystalline $CaSO_4 \cdot H_2O$ system is that mechanical-chemical coupling of deformation and hydration is central to permit water to reach the reaction zone and cause significant mechanical weakening. The stability of natural evaporites is of major interest in various settings, especially in contexts of underground structures with a variety of purposes, including road and tunnel construction and monitoring, mining of evaporites, and where caverns in evaporites are used as Geo-Energy storage facilities. In general, evaporitic rock salt deposits are anything but homogeneous or monomineralic (Stewart, 1963), with gypsum and anhydrite being two of the nine most important minerals. Organisations in Germany and the United States of America are already storing low- and intermediate level nuclear waste in repositories within rock salt deposits. The basic assumptions are that rock salt functions as a seal, with halokinesis 'healing' potential leaks. The need for more studies to determine the safety and efficiency of rock salt deposits is widely recognised.

The findings of this study, mechanical-chemical weakening through hydration of anhydrite along stressed fractures, show how rapidly mechanical weaknesses may form and threaten the stability of caverns in natural evaporite deposits. This needs to be included into future stability models. Anhydrite-bearing evaporite sequences are commonly the weakest horizons in sedimentary basins and form detachment horizons in foreland fold and thrust belts (e.g., Heard and Rubey, 1966; Hildyard et al., 2011). Hydration of the anhydrite through stressed fractures must further weaken the mechanical strength of such sequences and make the formation of detachment horizons easier.

The findings of this study also have implications for hydration in a wider variety of geological settings. The $CaSO_4 \cdot H_2O$ system could be seen as an analogue for other rock systems that are controlled by hydration, dehydration, and stress. Common fluid pathways in the Earth include faults, shear zones, and stratigraphic aquifers. The study suggests that hydration along such pathways can be rapid and generate planes of significant weakness under differential stress. Deep crustal earthquakes are often associated with local weakening of the generally dry, mechanically strong deep crust, through fluid-driven metamorphic reactions (Jamtveit et al., 2019). Studies from the Bergen Arcs in western Norway show that fluid migration through shear zones facilitates highly localized eclogitization of anhydrous (granulite) crust along these zones (e.g., Austrheim and Griffin, 1985; Austrheim, 1987; Jamtveit et al., 1990, Jamtveit et al., 2019) and can results in transient mechanical weakening, brittle deformation and earthquakes (e.g., Jamtveit et al., 2019; Bras et al., 2021). At an early stage, eclogite facies mineralogy is even known to be found as veins in extension fractures (Jamtveit et al., 1990). Subduction of oceanic and continental crust (Pérez-Gussinyé and Reston, 2001; Ranero et al., 2003; Bayrakci et al., 2016) transports water even to the deep mantle and creates local water rich horizons.

## 5 Conclusions

This is the first study that looks at the coupled mechanical behaviour and microstructural evolution during hydration in natural samples of anhydrite. Experimental hydration under non-hydrostatic stress conditions was successfully achieved over several hours and evidence was found for newly formed gypsum in post-experimental 'wet' mode and steady-state differential



compaction (SSDC) mode samples. Syn-experiment gypsum-filled veins and breccia veins with large gypsum clasts formed in extensional and shear orientations. Significant mechanical weakening of the natural Òdena anhydrite accompanied rapid hydration under non-hydrostatic stress conditions during SSDC mode experiments. The SSDC results in decreased (~ 14 to ~ 41 %) peak strength and lower differential stress and strain during failure compared to the 'wet' and dry mode tests. The mechanical-chemical link resulted in failure along gypsum veins after 71 minutes for one sample under SSDC conditions, whereas the other lasted ~ 6 hours in SSDC mode. EBSD analysis shows a selective topotactical relationship of large gypsum grains to the vein hosting anhydrite. The crystallographic orientations of the gypsum grains in new veins are also selective, systematic, and preferentially aligned parallel to the direction of maximum shear stress during the experiments. A model for the evolution of fracture formation and hydration involving mechanical-chemical coupling is proposed. The insights into rapid hydration under stress provided by this study have wider implications for geological and engineering settings.

**Data availability**

Data has been made available as supplements.

**Author contribution**

The experiments were conceptualised and designed by Johanna Heeb and David Healy, and carried out by Johanna Heeb. The Òdena rock samples were collected and selected by Enrique Gomez-Rivas. Experimental data were collected, processed, and analysed by Johanna Heeb. Microstructural data were collected, processed, and analysed by Johanna Heeb with assistance from Nicholas E. Timms. Johanna Heeb prepared the manuscript with contributions from all co-authors.

**Competing interests**

Co-author David Healy is a Topical Editor for EGU Solid Earth but was not involved in the journal review or editorial handling process for this manuscript. The authors Johanna Heeb, Nicholas E. Timms, and Enrique Gomez-Rivas declare that they have no conflict of interest.

**Acknowledgements**

Johanna Heeb acknowledges the Aberdeen-Curtin Alliance International Postgraduate Scholarship and a Curtin Publication Grant. The principal author thanks Prof. Chris Elders for supervision throughout this project and Prof. Thomas Blenkinsop, Prof. Steven M. Reddy, Prof. Mark Jessell, and Prof. Ian Alsop for their reviews of earlier versions of this work (in the form of a PhD thesis chapter). This study and its publication was supported by funding from the Natural Environment Research Council (NERC), grant number NE/T007826/1. Enrique Gomez-Rivas acknowledges the "Ramón y Cajal" fellowship



RYC2018-026335-I, funded by the Spanish Ministry of Science and Innovation (MCIN)/State Research Agency of Spain

(AEI)/European Regional Development Fund (ERDF)/10.13039/501100011033 and the DGICYT research project PID2020-118999GB-I00the, funded by the Spanish Ministry of Science and Innovation (MCIN)/State Research Agency of Spain (AEI)/10.13039/501100011033. We thank Juan Diego Martín-Martín and Federico Ortí for their advice and help with the Òdena Gypsum Formation sample collection and Roberto E. Rizzo for collecting the Volterra gypsum sample material in Italy.

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
