# Peer review of "Rapid hydration and weakening of anhydrite under stress: Implications for natural hydration in the Earth's crust and mantle"

_EGUsphere, 2023_

## Referee Comment (RC2)

Dear Editor,

As requested, I have reviewed the manuscript titled "Rapid hydration and weakening of anhydrite under stress: Implications for natural hydration in the Earth's crust and mantle" by Heeb et al., please find my general and specific comments below.

Heeb et al. present data from a series of deformation experiments run on anhydrite dominated samples that come from the Òdena Gypsum Formation and one reference experiment run on Volterra gypsum. The main result of the work is to show, for the first time, that a non-hydrostatic stress state influenced the hydration reaction both in timing and in extent. These results are then brought into a geological context and discussed to give the reader explicit understanding of why the experiments are meaningful.

**General comments:**
The contribution from Heeb et al. fills a gap in our understanding of deforming and reacting rocks, in particular hydrating rocks. The work is well written and the figures are generally very good at conveying the results with clarity. The science has been carefully conducted and is well detailed, which translates into clear results and a convincing narrative. It is great to see the recording of the threshold segmentation as has been done in supplementary material, it makes it very easy to assess visually what the authors have done. I particularly like the final discussion section and how it nicely captures a necessary extension of the model for décollement formation.

I have two minor comments that I would like the authors to address and one recommendation for future work:

- The first, is that the authors don't really discuss the alignment of the crystallography of gypsum with respect to the largest principal stress in detail. I find it a fascinating result that the planes in gypsum that contain the water molecules, {010}, form in the orientation of maximum shear stress. In the context of your mechanical-chemical coupling would you not want to discuss this further? These planes are also surely the weakest in the crystal structure, do you see any evidence of gypsum accommodating deformation along these planes? I know you discuss how gypsum in fractures ultimately act as locally weak regions for further shear fracturing, brecciation and eventual brittle failure, but I think you might want to make a stronger link to your crystallographic results that you have as they are probably pertinent to this argument. To be clear I am not suggesting more data are required, only that you think about linking what you already have to your existing text.
- My second comment is that you should add location data for your samples. You mention that they were collected from the field and others might want to replicate these experiments in future and collect similar samples. It would be useful to the long term reproducibility of your study to have these locations reported.
- My last comment, which I want to make clear is not something I think the authors should action in this work, is that I recommend that they move away from using simple thresholding. While their  bimodal histograms clearly justify the approach, fig. S18 highlights the limitation of the method. There are clearly regions that are grain boundaries that are being identified as gypsum that aren't. If the authors wish to use the area, or volume, values that they get to say something more quantitative about the reaction progress in future then I would recommend using more advanced segmentation methods as the uncertainties cannot really be referred to as 'minor' as the authors write in the text, largely because the uncertainties are not known. As a starter, check out the machine learning feature WEKA in Fiji. Regardless of this comment, I want to reiterate that I think that the authors are justified in their methods in the work under review and do an excellent job of recording and reporting their methods.

I have some specific comments below but otherwise congratulate Heeb et al. on an excellent piece of science that was a pleasure to read.

Best wishes,

James Gilgannon

**Specific comments:**

Line 99:
Two recent papers that stand out as missing for me here are Schrank et al. (2021) [https://doi.org/10.1038/s43246-021-00156-9] and Marti et al. (2020) [https://doi.org/10.1016/j.epsl.2020.116679].

Line 145:
'The anhydrite rocks have a minor natural gypsum content.'
Would you be able to give an estimate fraction? Even from previous studies if not from your own analysis. I think this would be good to report if you have it or access to it.

Line 241:
'…, viscous layer.'

I don't think this is the appropriate descriptive word for solid reaction products. For me viscous would only work if it was a liquid that you were describing. This description is also used on line 519 and I would change it there too.

Line 327:
'… compaction contrast …'

Do you mean that the regions have different amounts of compaction? And by this do you mean amount of porosity? I personally find compaction a confusing term here because you don't have access to the variable of compaction to compare. You only have microstructural descriptions like area of porosity.

Line 380:
'This suggests that there is an intrinsic link (or links) between the application of a non-hydrostatic stress field and the rate of the hydration reaction.'

I agree with Referee 1 (Sergio Llana-Funez) here that you would want to include a little more emphasis on permeability in your discussion. I say this because in a generic sense a low differential stress would not yield the results you have described because if the rock remained elastic and intact you would not have allowed as much access for water to allow hydration to proceed. Therefore, while a differential stress clearly has an effect, it must be partially through how it alters the microstructure of the rock. I am uncertain if one can claim that the link is differential stress -> rate of reaction from the results you present but rather, differential stress -> microstructural change -> change in permeability -> rate of reaction.

Line 517:
'The resulting lateral cheeks are either not faulted or extremely faulted, compared to the dry and 'wet' test samples.'

I found this a non-intuitive phrasing as I am not entirely sure what you mean by cheeks.

Figure 7 caption:

I might have missed it but I think the colours in b) aren't explained anywhere.

---

## Author Response (AR1)

**Rapid hydration and weakening of anhydrite under stress: Implications for natural hydration in the Earth's crust and mantle**

MS No.: egusphere-2023-161
MS type: Research article

**(1.1) Comments from Referee 1**

The manuscript entitled "Rapid hydration and weakening of anhydrite under stress: Implications for natural hydration in the Earth's crust and mantle" by Heeb and co-authors addresses the role of the hydration reaction in anhydrite to gypsum under effective pressure and differential stress to constrain the role of stress in the progress of the reaction and its rheological evolution. The experimental data set and particularly the microstructural characterization of starting and reactant products is outstanding.

I fully agree with the authors in the importance of hydration reactions in many several geodynamic scenarios, particularly in the context of active settings.

The main characteristic of hydration reactions is the increase in the volume of the solids. Although it is mentioned few times in the text, it is not given the relevance it has, as it probably explains the behavior of the reaction, particularly in the case of deforming reactant rock. The access of water to hydrate the rock requires the existence of a network of fluid pathways, which become sealed as the reaction progresses, given the large increase in volume of the solid during the reaction. In an static environment, this may be a contributing factor to halt the progression of the reaction. In the case of a dynamic environment, the continuous introduction of microfracturing, damage and pore space keeps the reaction going as it provides pathways for fluid to access unreacted material. The experiments done by Heeb et al. gain more relevance when addressed from this point of view. Perhaps it is no surprise that the reaction progresses faster or earlier in the case of a dynamic or stressed environment (thus strained) in comparison with a static set up. The experiments presented illustrate this very well.

But aside from considering this in the introduction and presentation of the tests, I also find several things in how the experiments are presented that need clarifying. In some cases the description of experimental procedure is confusing, in other perhaps more information is needed. The main issue is regarding the description of the tests, particularly the differences between the different sets of experiments, in some cases because there are several things mixed or because of the terminology used. For instance, the difference between "wet" and SSDC tests (steady-state differential compaction). Both require pore fluid pressure, the main difference resides in the magnitude of the effective pressure. But I'm not sure that this difference justifies their classification as different set, as there are other factors, mainly time. These also undergo large differences in time at pressure, the SSDC having almost a magnitude longer at stress that the "wet" ones. The graph in fig 3 shows the mechanical data in all tests, but it does not show the differences in confining pressure, and also in effective pressure, where there is a large difference between Ò1 and Ò7 or Ò8. It would be better for the reader to show separately the curves at different effective pressure, for instance, or color coding tests under similar conditions.

The usage of "initiating the strain rate" is somehow confusing. I take it to mean initiating the loading, which is what it is normally used. But one needs to bear in mind that one thing is the moving rate of the piston, which is easily kept constant, another is keeping the strain rate constant, which requires to recalculate the speed at which the piston moves to keep strain rate constant. I'm aware the differences may be minimal at low strain but they will increase as the shortening of the samples builds up. And by the look of some of the samples, we are probably closer to the latter than the former.

With regards to the procedure of the testing, in some of the experiments, there are instances where the piston is stopped, but the differential stress is kept constant, which is odd, because as soon as you stop the loading, the sample will start to relax reducing the differential stress. In relaxation tests, the strain rate will vary with time, in that case orders of magnitude depending on the times at which the sample is left to relax. How exactly is conducted this part of the experiments is not clear in the description, but in Fig 7 there is a segment in the loading curve where effective stress is kept constant while strain keeps accumulating, for several hours. The time is perhaps not too relevant for the amount of strain, but it certainly is for the progress of the reaction, since at this stage the sample remains stressed, thus, in conditions potentially favouring the progress of the reaction.

The name given to the tests Ò1 and Ò2 is confusing: "steady-state differential compaction under fluid pressure" (line 219). As written it can be understood as the fluid pressure what produces the compaction, when in fact it is the effective pressure which does. If the "strain rate" is put on hold, probably meaning that the piston is stopped, then the reaction is progressing during relaxation at high effective stress, in this case provided by the initial 100+ MPa differential stress, which will reduce very likely over time. The time at which the samples are left loaded is important (I assumed is tssdc in table 2) as despite some relaxation of stresses, they presumably will still be high enough to favour compaction.

However in line 260 it is said that the piston is stopped before fluid pressure was applied, is that correct?? If that is the case, then it is possible that a lot of fracturing is induced by reducing effective pressure. This part requires more detail. If it is how it is said, then certainly "steady-state" is not the right word to describe this type of experiment.

The microstructural work is outstanding and complements very well the lab work, once the procedures are presented more clearly. I would also suggest to rebrand the names of the tests to reflect more objectively the type of test.

Once the flow of events in the running of the tests is clear, I would probably put more emphasis on the fact that the time at high stress is the key factor to enhance further hydration.

I have some other minor comments that I make in order of appearance.

In the graphical abstract I wonder whether the first and last sketches are oriented similarly to the three middle sketches or that the shear sense in the fault in the sketch to the right is wrong, as it does not agree with the major faults in the middle three drawings. As far as one can tell, there are no reverse fault movements in the sample cylinder under vertical shortening.

Line 88, I would rewrite "… influencing the reaction activity and kinetics of hydration to anhydrite…"

Line 90, I would replace "material-specific characteristics (petrography)" by "microstructure"

Section 1.2 Mechanisms of anhydrite hydration

Somewhere in the text, the volume increase related to hydration needs to be given, perhaps this is the section. This is a key parameter and may potentially control the progress of the reaction.

3.1.2 Mechanical data

I would not consider in the stress-strain curves anything that happens before the linear elastic behavior as they are probably artifacts, more to do with the assembly of the sample and the adjustment of the loading column than with the internal deformation of the sample. And any of this can be seen in the graph in Fig. 3 anyway.

Line 280, That "gypsum infill implies an extensional component to the kinematics of these structures" very much depends on whether the net solid volume increase exceeds the porosity generated during fracturing. If not, then it is not strictly necessary. That is one reason why the volume increase during gypsification needs to be given. And consequently, an estimate of the porosity generated during fracturing (which perhaps will be easily estimated using the SEM micrographs or even EBSD images).

Fig. 5

The size of some of the gypsum grains is very large and will certainly biased the CPO when using the complete data set. Is it the same with 1 point per grain as used for anhydrite?

Lines 309-321

CPO data in anhydrite is shown as point per grain, while gypsum is the complete data set. I wonder how much that plays a role in the interpretation in this paragraph.

4.2.1. Rapid hydration of anhydrite under stress

Other than stress is the fact that there might be some microfractures produced due to high effective pressure, which is what ultimately allows the fluid to permeate the samples and have access to the inside of the specimens. This is not considered here and it is key, given that as it is said, there are fractures and grain size reduction by comminution.

In this regard, it is important to clarify whether the fluid pressure to SSDC tests is applied after loading to +100 MPa or before.

Line 385, I wouldn't use "quasi-elastic stress-strain behavior" to refer to the inelastic part of the curves.

Line 390, "re-application of strain rate": reloading of the sample from x differential stress to maximum differential stress.

Fig. 7

would indicate in the graph the amount of time at SSDC.I

Fig. 8.

If sigma 1 is vertical, then the shear sense in the sketch to the right (reverse fault) is wrong.

Line 468-470. That the weakening of sample Ò2 comes from the appearance of gypsum instead of the fracturing is difficult to ascertain from outside, really.

Line 474, A consequence of hydration under stress is the weakening of the sample during deformation.

Line 476, tests Ò4 and Ò7 are not at slow strain rates, but the highest according to table 2.

Lines 482-486, "A stronger connected shear fracture network developed until the onset of isotropic principal stress conditions…" If these experiments were initiated at high differential stress, I can't see how they are evolving at isotropic principal stress conditions .

Line 498, "Chemical potential depends on a 'weighted' mean stress, which means that the magnitude and orientation of stress have a relatively minor impact". Surely the magnitude is important, since influences the mean stress. Is this quote correct?

Line 546-549, ...” fluid migration through shear zones facilitated highly localized eclogitization of anhydrous (granulite) crust along these zones and can result in transient mechanical weakening, brittle deformation and earthquakes”.

I'm not sure this is an adequate analogue system for the gypsification . Gypsum is much weaker that anhydrite, I'm not so sure eclogite is that much weaker than granulite. Also, the weakening in the granulite case is the consequence of the fracturing, not necessarily of the hydration itself, unless the eclogites were deformed. If the latter is the case, it needs to be mentioned.

**(1.2) Author's response**

Comment from Author to Reviewer 1

We thank Reviewer 1 for their very thoughtful and knowledgeable comments. Based on the changes

due to raised issues we believe that the manuscript has improved significantly.

Kind regards,

Johanna Heeb

on behalf of the authors

General comments:

Comment about the importance of the increase in volume of solids as main characteristic of hydration

Thank you. Indeed. The key role of volume change on the reaction has been discussed in section 4.3 under the aspect of observed mechanical-chemical coupling.

Description of tests, terminology, classification as different sets

The classification as different tests ("wet" vs. former SSDC, now renamed as CSDC) is necessary. Yes, time (i.e., duration) plays an important role, as mentioned in the text and listed in table 2. CSDC also means that fluid pressure is introduced only after reaching ~100 MPa of differential stress. Therefore, the effective pressure changes significantly – i.e., abruptly. Additionally, the load is removed, and confining pressure held constant, which also justifies the difference in classification. We prefer to keep these categories for clarity.

Figure 3, improvement to show differences in confining pressure and effective pressure, color coding

Thank you. Figure 3 has been revised to better reflect the different test conditions. The names of the experiments have been changed to and a new colour code has been used to highlight similar conditions. To further highlight the differences in effective pressure, two stress strain diagrams have been made out of one. Differences of confining pressure and effective pressure are listed in table 2 as well.

The usage of "initiating the strain rate"

Thank you. The term was replaced by "initiating axial load" in the text when applicable.

Constant strain rate vs. speed of piston movement

Thank you for pointing this out. Yes, the moving rate of the piston was used. After the 'CSDC' mode (former SSDC) and with reapplication of the axial load via moving the piston at the same rate as at the beginning of the test, the strain rate should be different from before, because the length of the sample has decreased. Throughout the manuscript, when it applies, the term "strain rate" was replaced with "axial load" and "displacement rate". It was important to have dynamic displacement.

Description of CSDC tests – stress relaxation?

We maintained an external differential stress – i.e., the axial stress and confining pressure were kept constant in the phase under discussion. The strain (and strain rate) evolution can only be judged from the axial displacement transducers. With the load (axial and radial) staying constant, stress relaxation was not observed. It can be speculated that with time a relaxation effect would set in.

Change of using "steady-state differential compaction under fluid pressure" (line 219) to describe the testing mode

Agreed, the term "SSDC under fluid pressure" was used twice in the manuscript before. 'Under fluid pressure' has been deleted for both cases and SSDC has been changed to constant stress differential compaction (CSDC).

Relaxation and the factor of time for which the samples are left loaded

We have constant applied stress (an external boundary condition) and are not speculating on the internal deformation (relaxation) of the sample. Indeed, the time at which the samples are left loaded is listed in table 2 under $t_{CSDC}$.

Comment about line 260, stopping of piston movement, fracturing induced by reducing effective pressure, steady state being not the right term to describe this phase of the experiment.

It is correct that the piston was stopped before the fluid pressure was applied.

Thank you. We have no diagnostic criteria as to exactly determine when the fracturing happened (e.g., from acoustic emissions). All mentions of "steady state differential compaction" have been changed to "constant stress differential compaction" and "SSDC" has been changed to "CSDC" throughout.

Rebranding of names of the tests

Thank you. New names have been assigned to better reflect the test modes of individual samples. (H1, H2 for the CSDC tests, W1-4 for the 'wet' tests, and D1 and D2 for the dry tests).

Comment on emphasizing time as the key factor to enhance further hydration

Time (or timing) is a key factor of the discussion, emphasized in section 4.2.1 about the rapid hydration evident from the study. Additionally, there is the section 4.2.2 on the timing relations of hydration and spatial distribution. Section 4.3 on mechanical-chemical coupling mentions the amount of time of contact as a factor for the amount of hydration. It is also important to that the degree of hydration depends on other key factors as well, as it is important what happens in that 'time', for example with the space available, or nucleation rate, which can both control and stop hydration (see Discussion section).

**(1.3) Author's changes in manuscript**

Graphical Abstract:
Thank you, that is absolutely right. The arrows marking the shear sense in the right sketch have been changed to realistic direction, following the sketches in the middle.

Line 88:
Agreed, this is a very good adjustment that will be adopted.

Line 90
Done, with the addition of 'mineralogy' to include mineral content, as this is not included in the Earth Science definition of 'microstructure'

Section 1.2 Mechanisms of anhydrite hydration
Thank you. Volume change is referred to as 'swelling' and mentioned in the introduction with reference to 3 papers. This then links to section 4.3 Mechanical-chemical coupling. I have added to the sentence in the introduction (lines 83, 84), which reads now: "… and the complex expansion or swelling of theoretically up to 60 % volume increase associated with hydration …"

3.1.2 Mechanical data
Thank you for pointing this out. To address this issue, the following three sentences have been deleted: "All samples show an initial phase of rapid hardening up until approximately 10 to 20 MPa differential stress. After this, total strain either stabilises or shows a minimal increase, with increasing stress. The next stage is a phase of linear elastic deformation until yield stress is reached, after which the differential stress decreases."

Line 280
Thank you. An assessment has been done by analysing SEM images with ImageJ. Results can be found in the Supplement material in Fig. A16, A17, A18 and A19. Table A1 lists the % of black and white pixels. Estimation of porosity is biased by black pixels due to grain boundaries. Reviewer 2 made a comment addressing threshold analysis as method.

Fig. 5 & lines 309-321:
Yes, it is roughly the same. The pole figures of gypsum grains using 1 point per grain are included in the supplement material to this publication (Fig. A10), where X and Y axis are not aligned to the core orientation. The 1 point per grain dataset of gypsum contains 2385 data points, m.u.d. spans from 0.01 to 4.94 and displays clustering of data points at similar positions.

4.2.1. Rapid hydration of anhydrite under stress
Thank you. The missing clarification of application of fluid pressure during the CSDC tests has been addressed by adjustments of Figure 7.

Line 385
The sentence has been changed to "During the initial loading phase, the onset of intragranular fracturing concentrated in the centre of the core and the orientation of shear planes …". The last sentence in the paragraph has also been modified to address this comment and reads now as follows: "This suggests that there is an intrinsic link (or links) between the application of a non-hydrostatic (effective) stress field and the rate of the hydration reaction."

Line 390
Thank you, "re-application of strain rate" has been replaced by "reloading of the sample from SSDC differential stress"

Fig. 7
Agreed. The time has been added in the revised version of the figure.

Fig. 8.
Yes, very true. The figure has been adjusted accordingly.

Line 468-470

This sentence has been adjusted: "due to the appearance of mechanically weaker gypsum and dynamic opening and filling of cracks".

Line 474

Thank you. This has been adjusted and reads now: " A consequence of hydration under stress is the weakening of the sample during deformation."

Line 476

Thank you, yes there is something wrong. The strain rates do differ in speed, from fastest W1 (former Ò3) to slowest W2 and D1 (former Ò4, and 7). The sentence was changed as follows: "Faster strain rate (W1, $4.4 \cdot 10^{-5} s^{-1}$) and higher stress conditions (W2, $P_c$ = 100 MPa, $P_f$ = 90 MPa) generate weaker peak strengths."

Lines 482-486

Thank you. There is no stress relaxation, but indeed constant stress is maintained during this phase. The term has been changed to "constant axial and radial stress conditions" to address this.

Line 498

Thank you. The quote was checked and adjusted accordingly. The words 'relatively minor' have been deleted.

Line 546-549

The connection is that fluid migration can produce embrittlement and at the same time facilitates the reaction. Therefore, we find that the system can serve as an analogue for gypsification. Two sentences have been added according to the missing information provided by the reviewer: "However, weakening in the case of granulite is the consequence of fracturing, not necessarily of hydration, unless the eclogites were deformed." and "Although the difference in strength between granulite and eclogite is not as significant as that of anhydrite and gypsum."

**(2.1) Comments from Referee 2**

Dear Editor,

As requested, I have reviewed the manuscript titled "Rapid hydration and weakening of anhydrite under stress: Implications for natural hydration in the Earth's crust and mantle" by Heeb et al., please find my general and specific comments below.

Heeb et al. present data from a series of deformation experiments run on anhydrite dominated samples that come from the Òdena Gypsum Formation and one reference experiment run on Volterra gypsum. The main result of the work is to show, for the first time, that a non-hydrostatic stress state influenced the hydration reaction both in timing and in extent. These results are then brought into a geological context and discussed to give the reader explicit understanding of why the experiments are meaningful.

**General comments:**

The contribution from Heeb et al. fills a gap in our understanding of deforming and reacting rocks, in particular hydrating rocks. The work is well written and the figures are generally very good at conveying the results with clarity. The science has been carefully conducted and is well detailed, which translates into clear results and a convincing narrative. It is great to see the recording of the

threshold segmentation as has been done in supplementary material, it makes it very easy to assess visually what the authors have done. I particularly like the final discussion section and how it nicely captures a necessary extension of the model for décollement formation. I have two minor comments that I would like the authors to address and one recommendation for future work:

- The first, is that the authors don't really discuss the alignment of the crystallography of gypsum with respect to the largest principal stress in detail. I find it a fascinating result that the planes in gypsum that contain the water molecules, {010}, form in the orientation of maximum shear stress. In the context of your mechanical-chemical coupling would you not want to discuss this further? These planes are also surely the weakest in the crystal structure, do you see any evidence of gypsum accommodating deformation along these planes? I know you discuss how gypsum in fractures ultimately act as locally weak regions for further shear fracturing, brecciation and eventual brittle failure, but I think you might want to make a stronger link to your crystallographic results that you have as they are probably pertinent to this argument. To be clear I am not suggesting more data are required, only that you think about linking what you already have to your existing text.

- My second comment is that you should add location data for your samples. You mention that they were collected from the field and others might want to replicate these experiments in future and collect similar samples. It would be useful to the long term reproducibility of your study to have these locations reported.

- My last comment, which I want to make clear is not something I think the authors should action in this work, is that I recommend that they move away from using simple thresholding. While their bimodal histograms clearly justify the approach, fig. S18 highlights the limitation of the method. There are clearly regions that are grain boundaries that are being identified as gypsum that aren't. If the authors wish to use the area, or volume, values that they get to say something more quantitative about the reaction progress in future then I would recommend using more advanced segmentation methods as the uncertainties cannot really be referred to as 'minor' as the authors write in the text, largely because the uncertainties are not known. As a starter, check out the machine learning feature WEKA in Fiji. Regardless of this comment, I want to reiterate that I think that the authors are justified in their methods in the work under review and do an excellent job of recording and reporting their methods.

I have some specific comments below but otherwise congratulate Heeb et al. on an excellent piece of science that was a pleasure to read.

Best wishes,

James Gilgannon

**Specific comments:**

Line 99:

Two recent papers that stand out as missing for me here are Schrank et al. (2021) [https://doi.org/10.1038/s43246-021-00156-9] and Marti et al. (2020) [https://doi.org/10.1016/j.epsl.2020.116679].

Line 145:

'The anhydrite rocks have a minor natural gypsum content.' Would you be able to give an estimate fraction? Even from previous studies if not from your own analysis. I think this would be good to report if you have it or access to it.

Line 241:

'…, viscous layer.' I don't think this is the appropriate descriptive word for solid reaction products. For me viscous would only work if it was a liquid that you were describing. This description is also used on line 519 and I would change it there too.

Line 327:

'… compaction contrast …' Do you mean that the regions have different amounts of compaction? And by this do you mean amount of porosity? I personally find compaction a confusing term here because you don't have access to the variable of compaction to compare. You only have microstructural descriptions like area of porosity.

Line 380:

'This suggests that there is an intrinsic link (or links) between the application of a non-hydrostatic stress field and the rate of the hydration reaction.' I agree with Referee 1 (Sergio Llana-Funez) here that you would want to include a little more emphasis on permeability in your discussion. I say this because in a generic sense a low differential stress would not yield the results you have described because if the rock remained elastic and intact you would not have allowed as much access for water to allow hydration to proceed. Therefore, while a differential stress clearly has an effect, it must be partially through how it alters the microstructure of the rock. I am uncertain if one can claim that the link is differential stress -> rate of reaction from the results you present but rather, differential stress -> microstructural change -> change in permeability -> rate of reaction.

Line 517:

'The resulting lateral cheeks are either not faulted or extremely faulted, compared to the dry and 'wet' test samples.' I found this a non-intuitive phrasing as I am not entirely sure what you mean by cheeks.

Figure 7 caption:

I might have missed it but I think the colours in b) aren't explained anywhere.

**(2.2) Author's response**

Comment from Author to Reviewer 2

We thank Reviewer 2 for their very positive comments. We believe that the comments and changes based on them have completed and improved the manuscript and we will certainly use the suggested advanced segmentation methods for future work.

Kind regards,

Johanna Heeb

on behalf of the authors

General comments and recommendation

First comment on making a stronger link between crystallographic results and mechanical weakening: Thank you for noticing this, we agree. Changes have been made to make a stronger link between crystallographic results and mechanical weakening.

Thank you. The samples were collected by Enrique Gómez Rivas and Juan Diego Martín-Martín in the Igualada-Ódena area, where the Ódena gypsum member crops out along a line about 15 km long, trending NW-SW. It reaches a thickness of up to 30 m. The gypsum bed has three main terms: basal stromatolite, lower term and upper term. The lower term contains anhydrite masses up to several meters long, more abundant in the lower half. The samples were collected form this part of the formation, north of the municipality of Ódena.

Additional comment on emphasizing the permeability more during discussion, in agreement with RC1:

Thank you. The rate of change we see is due to the microcracking, which is driven by the differential (non-hydrostatic) stress. There are multiple factors that then control the reaction rate, i.e., volume change, ongoing microcracking, etc., this is part of section 4.

**(2.3) Author's changes in manuscript**

First comment on making a stronger link between crystallographic results and mechanical weakening:

The following sentences have been added to the section 4.2.3 where the crystallographic orientation of the newly formed gypsum is discussed:

" Gypsum has a monoclinic crystal structure, where a bilayer of water molecules, stacked along the b axis, separates bilayers of $Ca^{2+}$ cations and tetrahedral $SO_4^{2-}$ anionic groups. The adjoining layers are linked through weak hydrogen bonding, making the plane containing the water molecules (010) the weakest plane of shear, and causing the perfect cleavage of gypsum (Wooster, 1936). The two secondary cleavages (100) and (011) have much higher ultimate shear strength than any shear directions measured on (010) (Williams, 1988). The pole plots show that there is a strong preferred orientation of the (010) planes of the gypsum crystals parallel to the predicted shear fracture angle in the analysed area (Fig. 5), further favouring slip along the veins. The crystals in the veins are larger, longer that the aperture of the veins, which makes CPO likely to be a growth phenomenon rather than a result of deformation of pre-existing natural gypsum."

Second comment on the location of sample collection and reproducibility of the study:

A map with the location of the Ódena formation was added as supplement material (Fig. A20).

Line 99:
Agreed. The two suggested publications have been clearly missing and were added where suggested.

Line 145:
Thank you. A gypsum content of 10 to 15 % can be estimated for the original sample material, based on our data (included in the supplement material, Fig. A16, A17, A18, A19 and table A1). This has been added accordingly and the sentence has been changed to: 'The anhydrite rocks have a minor natural gypsum content of approximately 10 to 15 %.'

Line 241:

Indeed, this is about a liquid.

Line 327:
Agreed. 'Compaction' has been replaced with 'porosity'.

Line 380:
Thank you, this has been considered and addressed together with the corresponding comment of RC1. The following sentence has been added to section 4.2.1: "Microfractures produced during high effective pressure and further grain size reduction by comminution successfully facilitated hydration." And the last sentence of the section has been adjusted and reads now: "This suggests that there is an intrinsic link (or links) between the application of a non-hydrostatic stress field, microstructural change, change in permeability, and the rate of the hydration reaction."

Line 517:
Thank you. Lateral 'cheeks/bulges' has been replaced by 'chips' throughout.

Figure 7 captions:
Thank you. This figure has been adjusted accordingly.

**(3) Additional author's changes in manuscript**

Update of affiliations and email address for correspondence

Change of American English terms (crystallize, characterize, localize) to British English (crystallise, characterise, localise)

Figure 5: There is no panel e), so this has been deleted by the author(s)

Acknowledgements: Adjustments to include the two referees and acknowledge the support of the role of the MMF and JdLC at Curtin University and their funding.

References: Literature added by RC2 was added to the list of references.

Supplement A: Change of names in correspondence to manuscript.